# Development of a seroepidemiological tool for bat-borne and shrew-borne hantaviruses and its application using samples from Zambia

Rakiiya Sikatarii Sarii[1,2], Masahiro Kajihara[1,3], Zuoxing Wei[1,2], Sithumini M. W. Lokpathirage[1,2], Devinda S. Muthusinghe[1,2¤], Akina Mori-Kajihara[3], Katendi Changula[4], Yongjin Qiu[5], Joseph Ndebe[3,4], Bernard M. Hang'ombe[4], Fuka Kikuchi[6], Ai Hayashi[6], Motoi Suzuki[6], Hajime Kamiya[6], Satoru Arai[6], Ayato Takada[1,3], Kumiko Yoshimatsu[1,2]*

**1** Graduate School of Infectious Diseases, Hokkaido University, Sapporo, Japan, **2** Institute for Genetic Medicine, Laboratory for Animal Experimentation, Hokkaido University, Sapporo, Japan, **3** International Institute for Zoonosis Control, Hokkaido University, Sapporo, Japan, **4** The School of Veterinary Medicine, The University of Zambia, Lusaka, Zambia, **5** Faculty of Veterinary Medicine, Hokkaido University, Sapporo, Japan, **6** National Institute of Infectious Diseases, Tokyo, Japan

¤ Current address: National Research Center for the Control and Prevention of Infectious Diseases, Nagasaki University, Nagasaki, Japan
* yosimatu@igm.hokudai.ac.jp

**Data Availability Statement:** The authors confirm that all data underlying the findings are fully available without restriction. All relevant data are

## Abstract

### Background

Rodent-borne orthohantaviruses are the causative agents of hemorrhagic fever with renal syndrome and hantavirus pulmonary syndrome. Apart from the classic rodent-borne hantaviruses, numerous species of hantaviruses have been identified in shrews and bats; however, their antigenicity and pathogenicity are unknown. This study focused on developing a serological method to detect antibodies against bat- and shrew-borne hantaviruses.

### Methodology/Principal findings

Five bat-borne (Brno, Dakrong, Quezon, Robina, and Xuan Song) and 6 shrew-borne (Asama, Altai, Cao Bang, Nova, Seewis, and Thottapalayam) viruses were selected based on the phylogenetic differences in their N proteins. The recombinant N (rN) proteins of these viruses were expressed as antigens in Vero E6 and 293T cell lines using the pCAGGS/MCS vector. Antisera against the Nus-tagged rN fusion proteins of these viruses (mouse anti-Brno, Dakrong, Quezon, Robina, Xuan Song, Asama, Cao Bang, and Nova, while rabbit anti-Altai, Seewis and Thottapalayam) were also generated. Antigenic cross-reactivity was examined in antisera and rN-expressing Vero E6 cells. The rN proteins of almost all the tested viruses, except for the Quezon and Robina viruses, showed independent antigenicity. For serological screening of bat samples, 5 rNs of the bat-borne viruses were expressed together in a single transfection protocol. Similarly, 6 rNs of shrew-borne viruses were expressed. Reactivities of the mixed antigen system were also examined across the singly transfected Vero cell lines to ensure that all antigens were expressed. Using these antigens,

within the paper and its Supporting Information files.

**Funding:** This work was supported by the Science and Technology Research Partnership for Sustainable Development (SATREPS) (JP23jm0110019) received by KMY and AT and the Japan Agency for Medical Research and Development (JP24fk0108634) received by SA. URL: https://www.jst.go.jp/global/. The funders had no role in study design, data collection and analysis, decision to publish, or preparation of the manuscript.

**Competing interests:** The authors have declared that no competing interests exist.

bat serum samples collected from Zambia were screened using the indirect immunofluorescence antibody test (IFAT). Selected positive samples were individually tested for the respective antigens by IFAT and western blot assays using rN-expressing 293T cell lysates. Of the 1,764 bat serum samples tested, 11.4% and 17.4% were positive for bat and shrew mixed antigens, respectively. These samples showed positive reactions to the Brno, Dakrong, Quezon, Xuan Son, Robina, Asama, Altai, Cao Bang, or Thottapalayam virus antigens.

## Conclusions/Significance

These observations suggest that the mixed-antigen screening system is useful for serological screening For Orthohantavirus infections and that bats in Zambia are likely exposed to not only bat-borne hantaviruses but also to shrew-borne hantaviruses.

### Author summary

Advancements in scientific research tools in areas of emerging and re-emerging zoonotic diseases of public health importance are crucial to understanding the epidemiology of these diseases. Hantaviruses are not an exception. Hantavirus pulmonary syndrome (HPS) and hemorrhagic fever with renal syndrome (HFRS) are the two main rodent-borne diseases of public health importance. However, bat- and shrew-borne hantaviruses have also been reported with serological detection of Thottapalayam and Altai shrew-borne hantavirus infections in humans in Sri Lanka and Thailand. Based on these recent public health discoveries, antigenic development and evaluation are part of the key components in efforts put forward to understand the epidemiology of these viruses. This study provides researchers with the first antigenic comparison between selected bat- and shrew-borne hantaviruses using their rN proteins with their application in screening epidemiological samples (Bat sera) from Zambia. Hence, this comprehensive study can serve as the basis for further scientific collaboration within research communities.

## Introduction

Enveloped single-stranded negative-sense RNA viruses, known as hantaviruses, are members of the *Bunyavirales* order and the family *Hantaviridae*. Currently, a few rodent-borne orthohantaviruses are associated with 2 lethal illnesses in humans: hemorrhagic fever with renal syndrome which is mostly distributed in Europe and Asia, and hantavirus pulmonary syndrome (HPS) found in the Americas [1–3]. Orthohantaviruses are often transmitted to humans through the respiratory system via inhalation of aerosolized secretions or excretions [4,5]. Person-to-person transmission seldom occurs, with a few Andes virus-related cases of HPS in Argentina and Chile serving as exceptions [6–8]. The nucleocapsid (N) protein plays an important role in viral replication and assembly and serves as a serodiagnostic antigen [9–12]. Apart from in rodents, hantaviruses have also been detected in bats and shrews [13]. However, the potential for these emerging viruses to infect humans remains unclear.

With over 1400 species of bats (order Chiroptera) identified; bats constitute the second largest order of mammals. Bats are present on every continent except Antarctica. Most bats are insectivorous or frugivorous. Bats are natural reservoirs of many microbiological organisms,

and their social structure, lifespan, and flight facilitate the spread of zoonotic pathogens. Some bat-borne hantaviruses have been identified based on partial or complete S, M, and L genomic sequences [14]. In this study, viruses with full-length nucleocapsid (N) protein sequences were identified. They include Brno virus (loanvirus: BRNV) [15,16] isolated from the noctule bat, *Nyctalus noctula* (Czech Republic), Dakrong virus (mobatvirus: DKGV) [17] isolated from Stoliczka's trident bat, *Aselliscus stoliczkanus* (Vietnam), Quezon virus (mobatvirus: QZNV) [18] isolated from Geoffroy's rousette, *Rousettus amplexicaudatus* (Philippines), Robina virus (Orthohantavirus: ROBV) [19] isolated from the black fruit bat, *Pteropus alecto* (Australia), and Xuan Son virus (mobatvirus: XSV) [20] isolated from the ashy roundleaf bat, *Hipposideros cineraceus* (Vietnam). Phylogenetic analysis showed a close relationship between QZNV and ROBV viruses, similar to that between the DKGV and XSV viruses. In African countries, partial hantavirus sequences have been detected in bats captured in Sierra Leone [21], Cote d'Ivoire [22], and Gabon [23]. However, full genome sequencing and isolation of these viruses have not yet been performed.

The prototype of shrew-borne hantaviruses is the Thottapalayam virus (thottimovirus: TPMV), isolated from the Asian house shrew, *Suncus murinus* in India [24]. TPMV is thought to occur in a wide range of Asian countries, including China [25], Vietnam [26], Indonesia, and Thailand [27]. Shrews carry thottimovirus, orthohantaviruses, and other unclassified hantaviruses. Seewis virus (orthohantavirus: SWSV) is extensively dispersed in Europe and Siberia, and may naturally infect several species of Sorex shrews, such as *S. araneus*, *S. caecutiens*, *S. minutissimus* and *S. roboratus* [28]. Some related viruses have been reported in Kenya [29]. Sorex shrews are also carriers of Altai virus (ALTV), which is extensively dispersed throughout Russia, Mongolia, and Europe [28,30,31]. Asama virus (orthohantavirus: ASAV) [32] Nova virus (mobatvirus: NVAV) [33], and Cao Bang virus (orthohantavirus: CBNV) [34] have been detected in Japanese, European, and Chinese shrew moles (*Urotrichus talpoides*, *Talpa europaea*, and *Anourosorex squamipes*, respectively). In African countries, the shrew-borne hantaviruses Tanganya virus, Azagny virus, Kilimanjaro virus, and Ulugru virus have been reported in Crocidura and Myosrex shrews [22,31]. The shrew-borne viruses ALTV, CBNV, and ASAV are genetically distinct, and their classifications have not yet been approved.

Although many hantaviruses from bats and shrews have been reported, their antigenic differences have not been reported. Additionally, due to a lack of available tools, only a few small-scale studies have been reported on the potential for bat- and shrew-borne hantaviruses to infect humans. Cases of human seropositivity to TPMV, ALTV, and Imjin viruses have been reported [27,31,35]. In a previous study, we reported a diagnostic method using recombinant TPMV antigen [27]. We have also reported diagnostic methods for SWSV, ALTV, and ASAV and have shown low cross-reactivity among the N antigens of these viruses [27,36]. In this study, we expressed recombinant N (rN) proteins of 5 bat-borne hantaviruses (BRNV, DKGV, QZNV, ROBV, and XSV) and 6 shrew-borne hantaviruses (ASAV, ALTV, CBNV, NVAV, SWSV, and TPMV) for use in serological assays. Antisera to each rN protein were produced, and their cross-reactivities were examined. Systematic serological screening methods were established and applied to serum samples from bats captured in Zambia between 2006 and 2018.

## Methods

### Ethics statement

All animal experiments were approved by the Animal Studies Ethics Committee of Hokkaido University (24–0026). The mice were treated according to the laboratory animal control guidelines of the Hokkaido University Institutional Animal Care and Use Committee.

## Plasmids

The pCAGGS/MCS plasmid was used to express the rN protein of hantaviruses in mammalian cells [37]. For Nus-tagged rN expression in *Escherichia coli*, the pET43.1 system (Takara Bio, Japan; Novagen, Merck Millipore, USA) was used.

## Sequence information of N proteins of bat- and shrew-borne hantaviruses and construction of plasmids

The genetic differences among these viruses used in this study are shown on the left side of Fig 1A, and the nucleotide sequence information of the viruses are shown on the right. The open reading frames (ORFs) of the BRNV (KX845678), ROBV (MK165655), and CBNV (EF543524) genes were synthesized (Thermo Fisher Scientific, Life Technologies, USA). The ORFs of DKGV (MG663534), QZNV (KU950713), XSV (KC688335), and NVAV (KX512329) were amplified using PCR. ALTV, SWSV, ASAV, and TPMV N genes were previously prepared in both pCAGGS/MCS and pET43.1 plasmids [31,38]. Similarly, plasmids encoding the N proteins of BRNV, DKGV, QZNV, ROBV, XSV, CBNV, and NVAV were prepared to express rN proteins.

## Expression of rN proteins in Vero E6 cells for indirect immunofluorescence antibody test (IFAT) and in 293T cells for western blotting

An outline of the preparation and confirmation of the rN antigens is shown in Fig 1B. Vero E6 cells expressing the respective rNPs were prepared and seeded onto 24-well glass slides for indirect immunofluorescence antibody testing (IFAT). Briefly, Vero E6 cells grown in a 6-well plate were transfected with 2 μg of the expression plasmid vector pCAGGS/MCS prepared as described above using the TransIT-LT1 transfection reagent (Mirus Bio LLC, USA) according to the manufacturer's instructions. Twenty-four hours after transfection, the cells were collected by trypsinization, seeded onto 24-well glass slides (Matsunami glass IND.,LTD., Japan), and incubated for 24 hours in a 5% $CO_2$ incubator at 37˚C. The glass slides were washed with phosphate-buffered saline (PBS) and fixed with acetone for 10 min. After fixation, slides were briefly washed with distilled water and dried. The slides were stored at –80˚C until use. Similarly, 293T cells were transfected and collected 2 days after transfection and treated with 500 μL of 2× sample buffer (Fujifilm Wako pure chemical corporation) for western blotting. IFAT positivity was graded as +, ++, or +++ based on the intensity and number of positive cells. Triplicates of each sample were applied on 3x8, 24-well glass slides for all IFA analyses.

## Preparation of antisera to rN proteins

Rabbit anti-TPMV, anti-SWSV, and ALTV N proteins were prepared as previously described [27,31]. Similarly, rN proteins fused with Nus and histidine-tagged proteins at the N-terminal end were expressed using the pET43.1 system in BL-21 *E. coli* cell lines (Takara Bio, Japan; Novagen, Merck Millipore, USA) and purified using His Trap HP columns (Cytiva, USA) according to the manufacturer's instructions. We estimated the molecular weights of the purified antigens by western blotting using a mouse anti-NUS•Tag monoclonal antibody (Takara Bio, Japan; Novagen, Merck Millipore, USA). The anti-NUS antibody was used to confirm the preliminary expression of the various proteins in BL-21 cells before proceeding to produce the respective antisera by using HRP-conjugated anti-mouse (Jackson ImmunoResearch, USA) as secondary antibody. Purified rN antigens of BRNV, DKGV, QZNV, ROBV, XSV, ASAV, CBNV, and NVAV were used to immunize mice with a total dose of 120 mg (60, 30, and 30 mg) at 3-week intervals with Zeno particle adjuvant (Zenogen Pharma Co., Ltd. Japan) for the

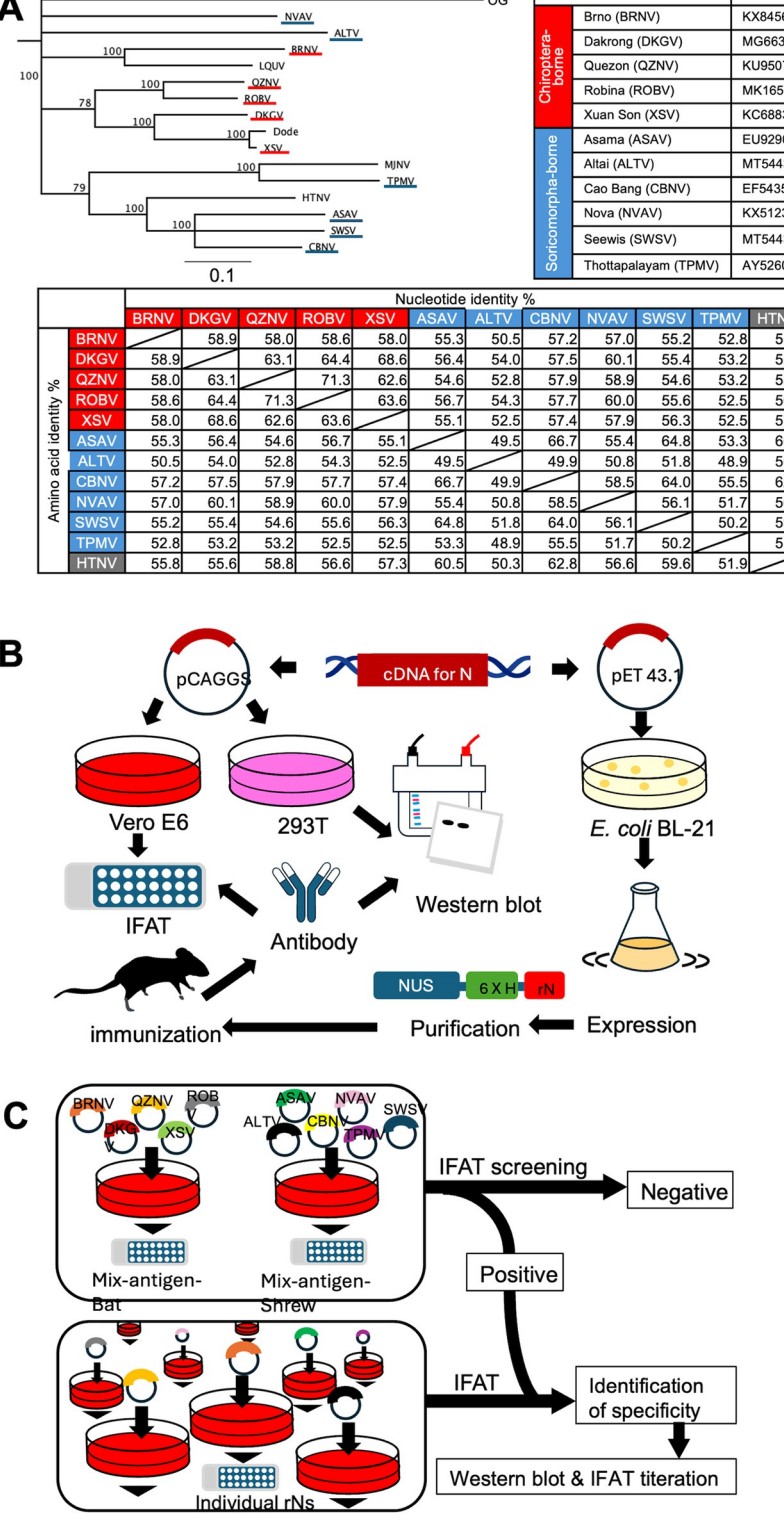

**Fig 1. Outline of experiments.** (A) Bat- and shrew-borne hantaviruses used in this study. The relationships between these viruses were determined by phylogenetic analysis based on the deduced amino acid sequences of their N proteins. The phylogenetic tree was constructed by neighbor-joining using the MAFFT alignment tool in Geneious Prime 2024 (MDF; Tokyo, Japan). HTNV; Hantaan orthohantavirus strain 76118 (M14626), MJNV; Imujin thottimovirus (EF641804). LQUV, Longquan loanvirus (JX465420). Other sequences are listed in the table. In the

phylogenetic tree, the names of bat-borne viruses are underlined in red and shrew-borne viruses are underlined in blue. The lower table shows the pairwise nucleotide and amino acid identity. (B) Expression of rNs as IFAT and western blot antigens. Further expression and purification of rNs as an antigen to prepare antiserum. (C): Strategy of screening bat serum samples. mixed-antigen-bat and mixed-antigen-shrew were used for the initial screening. Positive serum samples were then examined using individual antigens.

first 2 doses to produce the respective antisera. Alexa Fluor 488-labeled goat anti-mouse or anti-rabbit IgG antibodies (Thermo Fisher Scientific, Life Technologies, USA) were used as secondary antibodies for mouse and rabbit antisera. Subsequently, individual homologous antisera titration was performed and cross-reactivity between rN was assessed. Also, in this study, we investigated the reactivity of an E5/G6 monoclonal antibody against the expressed bat- and shrew-borne rN proteins.

## Novel screening tool for bat and shrew-borne hantaviruses

In this study, we developed a serological platform for screening antibodies against to 5 bat- and 6 shrew-borne hantaviruses using their rN protein antigens. Vero E6 cells were co-transfected with the mixture of pCAGGS/MCS containing N protein genes of the selected bat-borne (BRNV-DKGV-QZNV-ROBV-XSV) and shrew-borne (ASAV-ALTV-CBNV-N-VAV-SWSV-TPMV) viruses using the TransIT-LT1 transfection reagent (Fig 1C). Vero E6 cells grown in 6-well plates were transfected with the plasmids (0.5 μg of each plasmid of bat-borne virus and 0.4 μg of each plasmid of shrew-borne virus). Transfected cells were seeded onto 24-well glass slides as described above and evaluated using IFAT.

## Secondary antibody anti-bat IgG

The reactivity of horseradish peroxidase (HRP)-conjugated Protein A and G (Thermo Fisher Scientific, Life Technologies, USA) and HRP-conjugated Goat anti-Bat IgG (H+L) (Bethyl Laboratories. Inc., USA) were compared using ELISA as previously described [39]. Serially diluted sera from three bat species (*Macronycteris vittatus, Eidolon helvum, and Rousettus aegyptiacus*) were coated onto an ELISA plate and blocked. Respective HRP-conjugated proteins were used as secondary antibodies and OPD substrates for the detection. Anti-bat IgG was more reactive against the Bat IgG from the three species tested. (S1 Fig); hence, we purchased unlabeled Goat anti-Bat IgG (H+L) and 0.5 mg the IgG fraction was conjugated with Alexa 488 dye using an Alexa Fluor 488 protein labeling kit (Invitrogen, Thermo Fisher Scientific, USA). The conjugated anti-Bat IgG served as a secondary antibody for IFAT screening of bat serum samples at a dilution of 1:1000.

## Bat serum samples

Archived serum samples collected from 1,764 bats in Zambia were used for serological screening. These samples were collected from 2006 to 2018 with the permission of the Department of National Parks and Wildlife, Ministry of Tourism and Arts, Zambia (DNPW8/27/1) and the ethical approval of the University of Zambia Biomedical Research Ethics Committee (1382–2020). IFAT screening of all serum samples was performed using the co-transfected system at a serum dilution of 1:300. Next, IFAT using individually expressed antigens was used to identify bound antigens. Titration of the selected serum samples was performed, considering the reaction pattern and cross-reactivity. Some selected positive serum samples were further confirmed for reactivity by western blot assay, as described below.

## Western blotting

Similarly, 293T cells were transfected and collected 2 days after transfection and treated with 500 μl of 2× sample buffer (Fujifilm Wako Pure Chemical Corporation) for sample preparation for sodium dodecyl-sulfate polyacrylamide gel electrophoresis (SDS-PAGE).

Western blot assay was performed as previously described [31]. Briefly, 10 μL of cell lysate treated with SDS-sample buffer prepared as above were applied per lane. Proteins embedded into gels were then transferred to nitrocellulose membrane (Immobilon-P/IPVH, 0.45 μm, Sigma Aldrich) Rabbit and mouse immune serum (1:1000) and bat serum (1:500) were applied and incubated for 1 hour. Bound antibody was detected using secondary antibodies, HRP-conjugated anti-rabbit/HRP-conjugated anti-mouse (Jackson), HRP-conjugated goat anti-bat (Bethyl) IgG (H+L), and Amersham ECL Prime Western Blotting Detection Reagent/detection machine: Image Quant LAS 4000 mini (Cytiva, USA). All secondary antibody dilutions were 1:5000.

## Statistics

Fisher's exact test was performed using R software (R Foundation for Statistical Computing, Vienna, Austria) [40]. Relations between seropositivity were also examined by correlational analysis of the pairwise method on DATAtab [41] and the correlation ratio was examined using the Microsoft Excel function "CORREL" (Microsoft Corporation, USA).

## Results

### Expression of rNs in Vero E6 cells and their antigenic similarity

**Expression of rN antigens as IFAT antigens and western blot assay.**   To establish a serological screening assay based on IFAT, the rN proteins of hantaviruses were expressed in Vero E6 cells by transfection with mammalian cell expression vectors encoding N gene sequences. The expression of rN proteins was confirmed using homologous mouse or rabbit antiserum against the respective rN antigens. The antisera showed specific fluorescence signals as shown in Fig 2A. Diffused fluorescence in the cytoplasm and the presence of non-reactive cells indicated specific reactions of the antisera to rN-expressing cells. IFAT demonstrated that the transfection efficiency of each plasmid vector was approximately 10% to 20% (The total expression in three wells was compared to the unexpressed cells in the same well. Hence, the population of expressed cells was estimated at 15±5%). Similarly, the expression of individual rN antigens in pET43.1 as well 293T cells was confirmed by western blot analysis (Fig 2B). Strong signals from the rN-antisera reaction were detected at approximately 105kDa for NUS-tag fused antigens and 50 kDa respectively, which is the expected molecular weight of the hantavirus N protein.

**Evaluation of cross-reactivity.**   Confirmation of the individual rN protein was followed by homologous antibody titration testing for cross-reactivity between rN proteins (Table 1). No cross-reactivity was detected between these antigens and antibodies, except between QZNV and ROBV if 51200 was set as tentative cut-off. However, there was almost one-way cross-reactivity between the anti-ROBV antibody and QZNV antigen. Similarly, one-way cross-reactivity was observed between SWSV and CBNV.

### Confirmation of simultaneous antigen expression of rNs of 5 bat-borne and 6 shrew-borne viruses

For serological screening using IFAT, simultaneous antigen expression of the rNs of 5 bat-borne viruses (mixed-antigen-bat) and 6 shrew-borne viruses (mixed-antigen-shrew) was

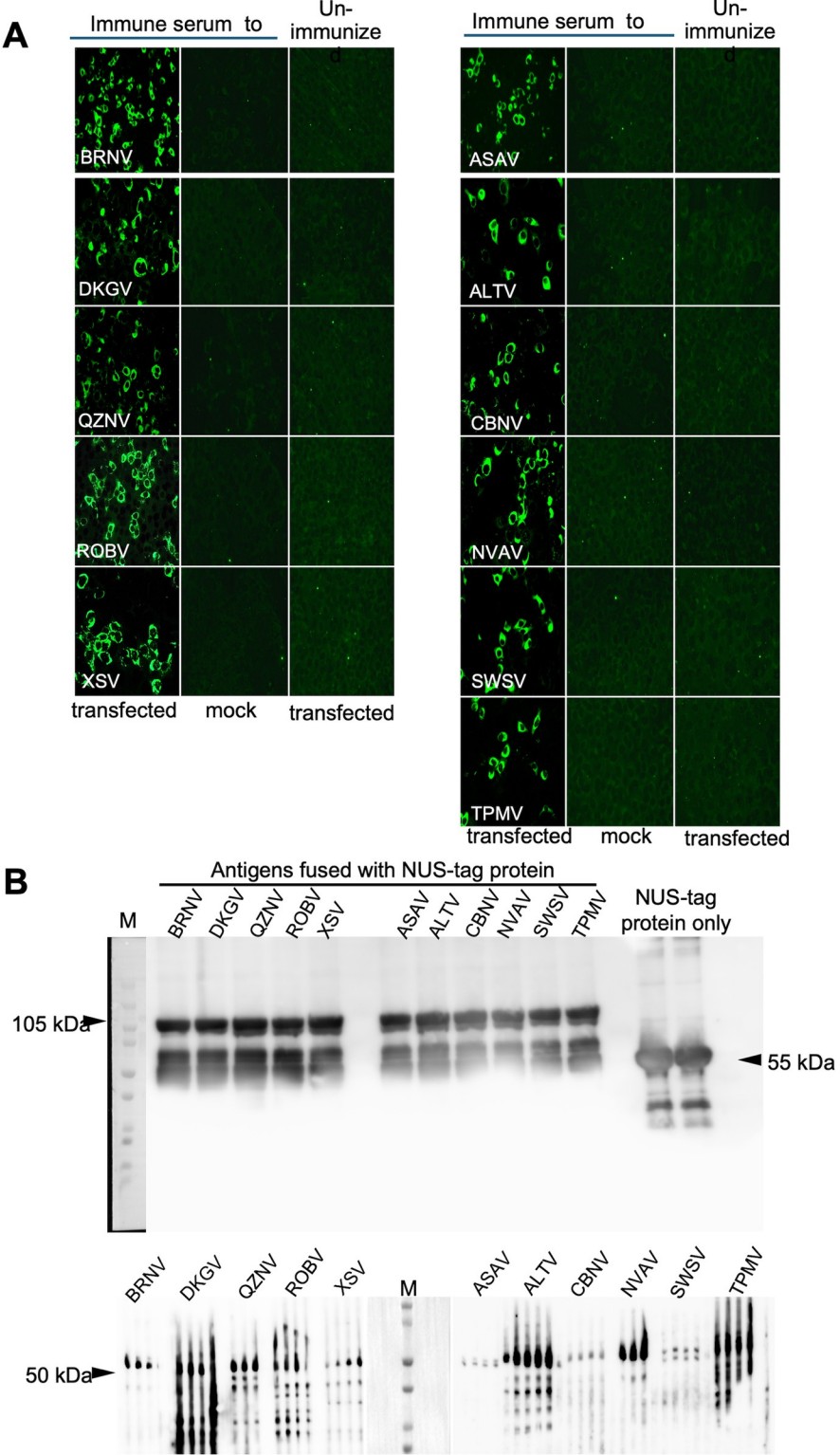

**Fig 2. Expression of individual rN proteins of bat- and shrew-borne hantaviruses.** (A) Vero E6 cells expressing individual antigens on indirect fluorescent antibody testing (IFAT). Mouse and rabbit antisera were used at dilution 1:1000. Anti-mouse and anti-rabbit IgG Alexa Fluor 488 were used as secondary antibodies for the mouse and rabbit antisera, respectively. The negative control contained Mock transfected Vero E6 cells with immunized and transfected cells with unimmunized serum respectively. (B) Individual rN protein blots with molecular weight markers. The upper

panel shows the estimated blot size 105 kDa for the individual antigen fused with NUS-tag protein. Mouse anti-Nus•Tag monoclonal antibody (Takara; Novagen) was used for the detection of NUS-tagged protein. The lower panel shows the estimated molecular weight of rN expressed in 293T cells those are approximately 50 kDa. Antisera were diluted 1:1000 and applied to the 4 slits of Screener Blotter 28 (Sanplatec, Osaka, Japan). Anti-mouse and anti-rabbit IgG HRP conjugate s were used as secondary antibodies for the mouse and rabbit antisera, respectively.

performed. Antigen expression was confirmed by IFAT using homologous antisera (Fig 3). Vero E6 cells without transfection did not react to any of the antisera.

## Seroprevalence of hantavirus infection in bats captured in Zambia

**Serological screening with mixed-antigen-bat and mixed-antigen-shrew.** Of the 1,764 bat serum samples collected from the 11 bat species, 201 and 307 were positive for mixed-antigen-bat and mixed-antigen-shrew, respectively, exhibiting various intensities of fluorescence at 1:300 dilution (Table 2). Of the 508 total positives, 107 were positive for both antigens and 401 serum samples were positive for mixed-antigen-bat or mixed-antigen-shrew, and these were further examined by IFAT using individual antigens.

**Typical reaction patterns of bat serum in IFAT and western blot.** Generally, the reaction patterns of the bat serum sample on IFAT and WB showed varying degrees of reactivity to the respective rN proteins. For example, ZFB06-11 (ZFB = Zambian fruit bat, 06 = year, 11 = sample number) showed reactivity to DKGV and XSV antigens on individual IFAT but was reactive to XSV only on western blotting assay. Similarly, ZFB14-151 showed reactivity to both BRNV and ROBV on individual IFAT; however, only BRNV was reactive on western blotting. Other reaction patterns indicated mixed infection, as seen in ZFB11-40, where individual antigens showed reactivity to both BRNV and CBNV antigens and were detected on western blotting assays. Excerpts of the typical reaction patterns of bat serum using IFAT and western blot are shown in Figs 4 and 5. IFAT antibody titers of these sera were shown in Table 3.

**Individual antigen screening.** As shown above, bat serum samples positive to mixed-antigen-bat and mixed-antigen-shrew were examined for reactivity to individual rN antigens. Of the 201 samples positive for the mixed-antigen-bat, 97, 15, 80, 38, and 41 samples reacted with the rN proteins of BRNV, DKGV, QZNV, ROBV, and XSV, respectively. Of the 307 samples

**Table 1. Cross-reactivities of antisera to rNs of bat- and shrew-borne hantaviruses.**

| Recombinant N antigen of | | IFA titers of polyclonal antisera to rN of | | | | | | | | | | |
|---|---|---|---|---|---|---|---|---|---|---|---|---|
| | | BRNV | DAGV | QZNV | ROBV | XSV | ASAV | ALTV | CBNV | NVAV | SWSV | TPMV |
| Chiroptera-borne | BRNV | 204800 | - | - | - | - | - | - | - | - | 3200 | - |
| | DAGV | - | 102400 | - | - | - | - | - | - | - | - | - |
| | QZNV | - | - | 204800 | 51200 | 3200 | - | - | - | - | - | - |
| | ROBV | - | - | 6400 | 204800 | 3200 | - | - | - | - | 3200 | - |
| | XSV | - | 3200 | - | - | 204800 | - | - | - | - | - | - |
| Soricomorpha-borne | ASAV | - | - | - | - | - | 102400 | - | - | 3200 | - | - |
| | ALTV | - | - | - | - | - | - | 204800 | - | - | - | - |
| | CBNV | - | - | - | - | - | - | - | 204800 | 3200 | - | - |
| | NVAV | - | - | - | - | - | 3200 | - | - | 102400 | - | - |
| | SWSV | - | - | - | - | - | - | - | 25600 | - | 204800 | - |
| | TPMV | - | - | - | - | - | - | - | - | - | - | 102400 |

-; <100

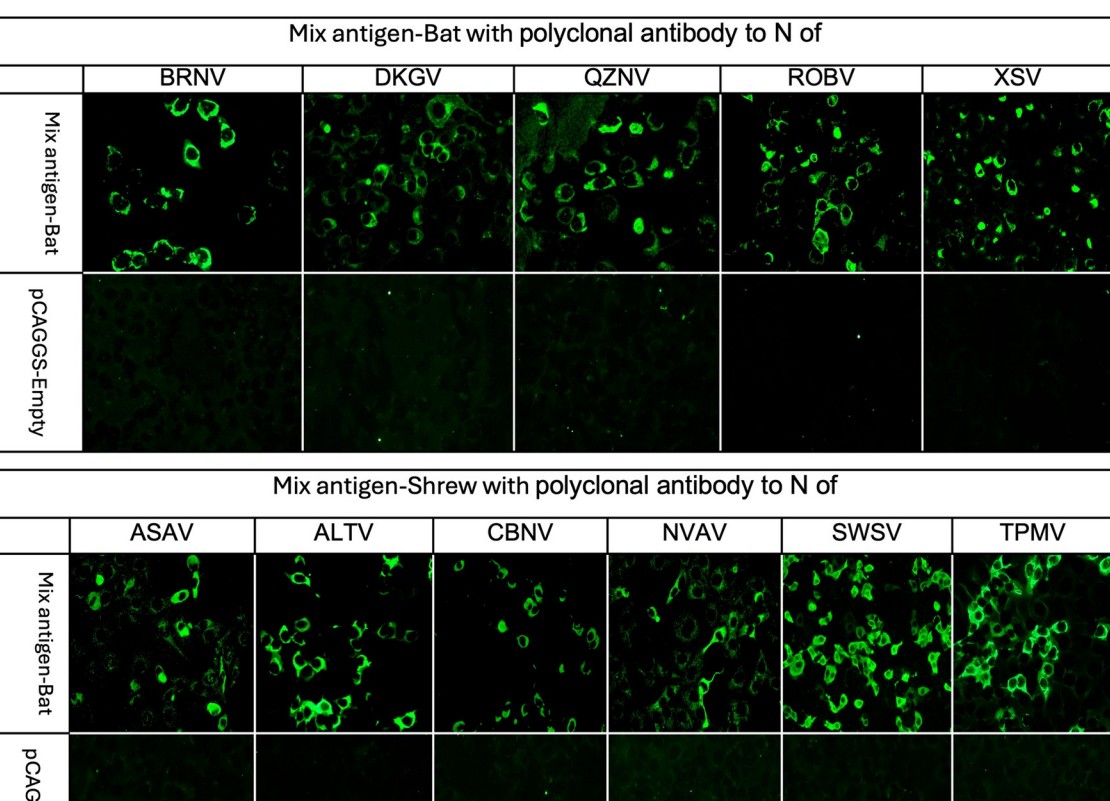

**Fig 3. Expression of each antigen in mixed-antigen-bat and mixed-antigen-shrew.** Vero E6 cells expressing multiple rN proteins were incubated with respective antisera to confirm the expression of each rN protein. All polyclonal antibodies to rN of BRNV, DKGV, QZNV, ROBV, and XSV for mixed-antigen-bat and ASAV, CBNV, NVAV, ALTV, SWSV, and TPMV for mixed-antigen-shrew at a 1:1000 dilution provided strong fluorescent signals, indicating the expression of each rN protein. In contrast, all antisera to rN were not bound to Vero E6 cells with empty pCAGGS plasmid DNA transfection.

positive for the mixed-antigen shrew, 90, 90, 48, and 86 samples reacted to the rN proteins of ASAV, ALTV, CBNV, and TPMV, respectively. None of the bat serum samples reacted to the NVAV and SWSV antigens (Table 4).

## Statistical analysis

A comparison of the seropositivity rates for mixed-antigen-bat and mixed-antigen-shrew antigens is shown in Fig 6A. Serum samples from *Rousettus aegypticus* showed significantly low seropositivity to mixed-antigen-bat, according to the Fisher's exact test. Three fruit bat species (*Ei. helvum*, *Epomophorus crypturus* and *R. aegyptiacus*) showed higher seropositivity rates than those of the insectivorous bats *Macronycteris vittatus* and *Nycteris thebaica*. Samples from other bat species were not analyzed because of the low number of samples. Next, the seropositivity rates of individual antigens within each species were compared (Fig 6B). The reaction profiles of fruit bats to the antigens of bat-borne hantaviruses were similar as seen *Eidolon helvum* with BRNV recording 7.9% (78) as the highest and 6.6% (65) for QZNV as the second highest. A similar pattern was observed in the *Rousettus aegyptiacus* species, as shown in Fig 6B. In contrast, reactivity to the antigens of the shrew-borne hantaviruses of *R. aegypticus*

Table 2. Summary of Mix-antigen-Bat and Mix-antigen-Shrew screening by IFAT for bat sera from Zambia.

| Species | Tested | Mix-antigen-Bat | | | | Mix-antigen-Shrew | | | | Specificities | | | | |
|---|---|---|---|---|---|---|---|---|---|---|---|---|---|---|
| | | Number of positives | IFA intensity | | | Number of positives | IFA intensity | | | Negative to both antigens | Positives to | | | |
| | | | + | ++ | +++ | | + | ++ | +++ | | either antigen | classification | | |
| | | | | | | | | | | | | both antigens | only bat antigen | only shrew antigen |
| *Eidolon helvum* | 986 | 156 | 100 | 39 | 17 | 195 | 109 | 48 | 38 | 716 | 270 | 81 | 75 | 114 |
| *Epomophorus crypturus* | 84 | 7 | 7 | 0 | 0 | 14 | 11 | 3 | 0 | 66 | 18 | 3 | 4 | 11 |
| *Hipposideros caffer* | 2 | 0 | 0 | 0 | 0 | 0 | 0 | 0 | 0 | 2 | 0 | 0 | 0 | 0 |
| *Macronycteris vittatus* | 182 | 0 | 0 | 0 | 0 | 3 | 2 | 0 | 1 | 179 | 3 | 0 | 0 | 3 |
| *Miniopterus* sp. | 38 | 0 | 0 | 0 | 0 | 0 | 0 | 0 | 0 | 38 | 0 | 0 | 0 | 0 |
| *Nycteris thebaica* | 93 | 4 | 3 | 1 | 0 | 1 | 1 | 0 | 0 | 89 | 4 | 1 | 3 | 0 |
| *Rhinolophus blasii* | 1 | 0 | 0 | 0 | 0 | 0 | 0 | 0 | 0 | 1 | 0 | 0 | 0 | 0 |
| *Rhinolophus swinnyi.* | 4 | 0 | 0 | 0 | 0 | 0 | 0 | 0 | 0 | 4 | 0 | 0 | 0 | 0 |
| *Rhinolophus* sp. | 4 | 0 | 0 | 0 | 0 | 0 | 0 | 0 | 0 | 4 | 0 | 0 | 0 | 0 |
| *Rousettus aegyptiacus* | 370 | 34 | 22 | 9 | 3 | 94 | 46 | 24 | 24 | 264 | 106 | 22 | 12 | 72 |
| Grand Total | 1764 | 201 (11.4%) | 132 | 49 | 20 | 307 (17.4%) | 169 | 75 | 63 | 1363 (77.3%) | 401 (22.7%) | 107 (6.1%) | 94 (5.0%) | 200 (11.3%) |

The degrees of reaction at 1:300 dilution were categorized as +, ++ and +++ respectively for Mix-antigen-Bat and -Shrew. This classification was done subjectively by two examiners because objective assessment was difficult. Therefore, 401 sera positive to either antigen were forwarded to further examination.

showed a significantly lower seropositivity rate to the CBNV antigen recording 1 positive out of 370 sera. In addition, a significant difference was detected between the seropositivity rates of the ASAV 6.2% (23) and ALTV 11% (41) antigens. Due to the inherent issues of non-qual sample sizes, comparisons between bat species were not recommended. The secondary antibodies used in this study were not guaranteed to react equally with all bat species tested; therefore, no cross-species comparative analyses were performed.

## Relationship between multiple reactivities

Next, cross-reactivities to the antigens of bat- and shrew-borne hantaviruses were examined using bat serum reactivity. The relationships between the number of samples testing positive to individual antigens were examined (S1A Table). Next, bat serum samples showing multiple reactivities to individual antigens were extracted. A total of 138 bat serum samples were analyzed, and the relationship between the respective positive reactions is shown in S1B Table. No significant relationship between the seropositivity to any of the antigen combinations was observed. These results indicate that no cross-reactivity between the antigens used in this study was detected in the bat serum samples.

## Discussion

Although the detection of bat- and shrew-borne hantaviruses has been increasing, serological screening assays for both bat- and shrew-borne hantavirus infections are not well established [11,27]. Additionally, few studies have assessed the antigenicity of bat- and shrew-borne hantaviruses. In this study, we developed serological screening assays for these viruses and studied the antigenic similarities between representative bat- and shrew-borne hantaviruses.

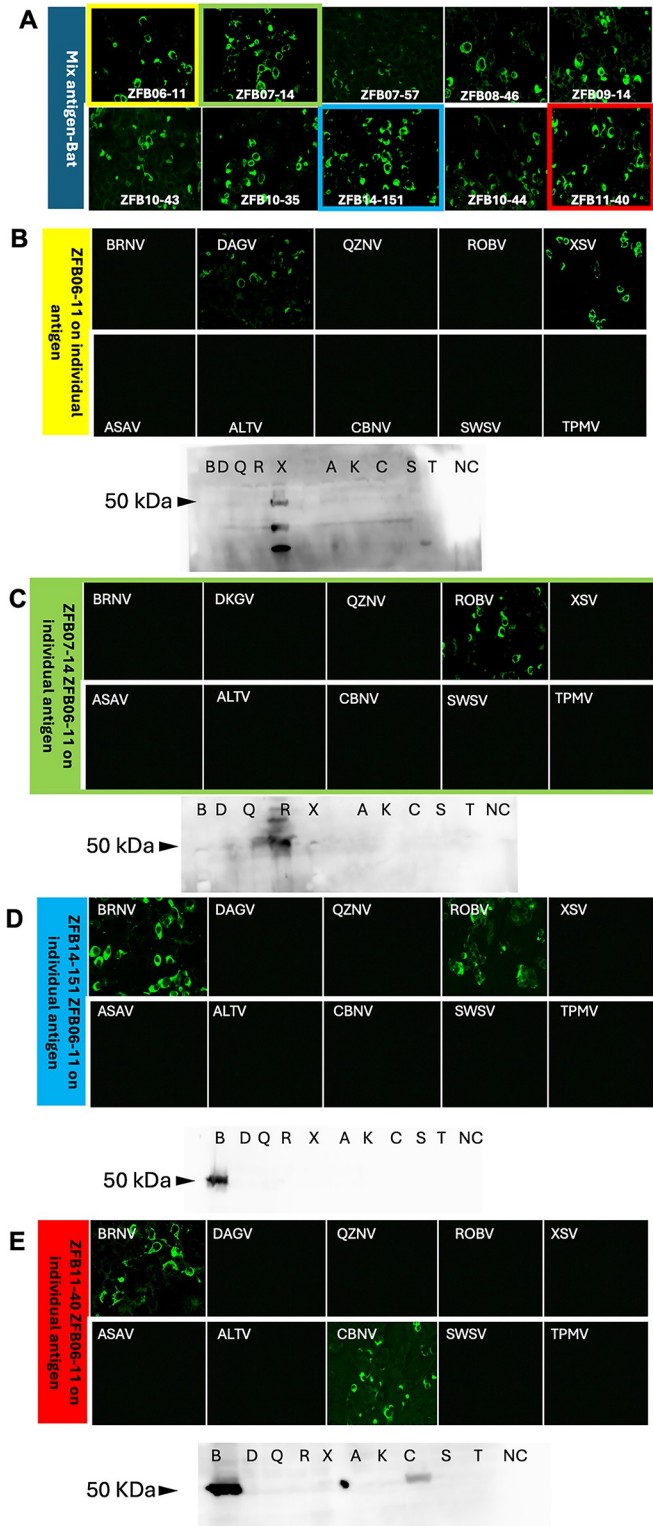

**Fig 4. Typical reaction pattern of bat serum samples to mixed-antigen-bat and individual antigen.** (A) Typical indirect fluorescent antibody (IFA) patterns of 10 selected bat serum samples for mixed-antigen-bat. (B) Immune reaction profile of ZFB06-11 bat serum in IFAT and western blot analyses. The IFA titers for the individual antigens are shown in the table. (C) IFA and western blot profiles of ZFB07-14 bat serum to individual antigens. (D) IFA and western blot profiles of ZFB14-151 bat serum to individual antigens. (E): IFA and western blot profile of–ZFB11-40 bat serum to individual antigens.

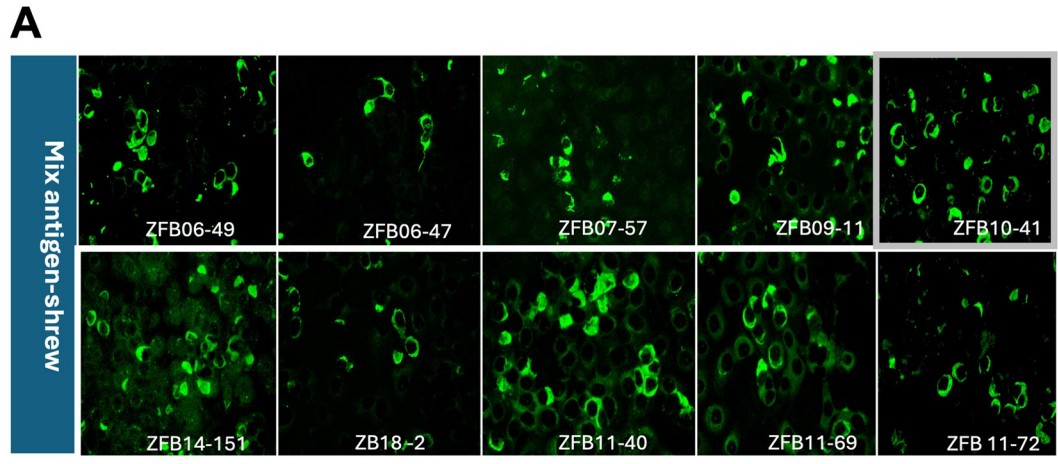

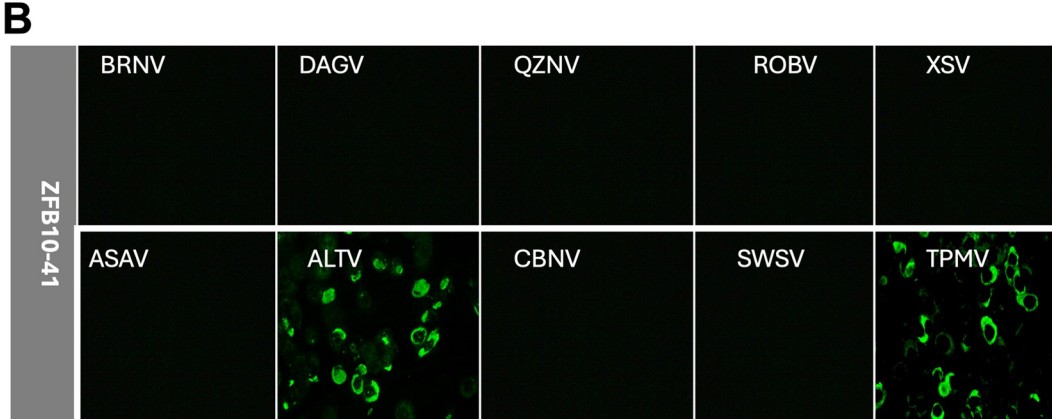

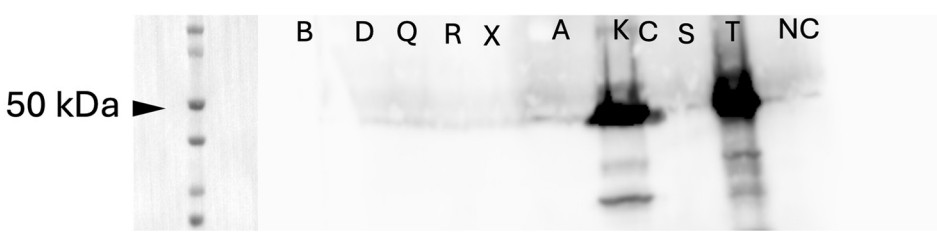

**Fig 5. Typical reaction pattern of bat serum samples to mixed-antigen-shrew and individual antigen.** (A) Typical indirect fluorescent antibody (IFA) patterns of 10 selected bat serum samples for mixed-antigen-shrew. (B) IFA and western blot profiles of ZFB10-41 bat serum to individual antigens. ZFB10-41 showed a positive reaction to both ALTV and TPMV and showed a strong positive reaction in the western blot assay.

Serological screening of frugivorous and insectivorous bats in Zambia was conducted using the established assays.

Although few reports on the antigenicity of the N protein of bat-borne and shrew-borne hantaviruses have been published, epitopes on the N proteins of bat-borne hantaviruses have been analyzed only *in silico* and common antigenic sites have been proposed [42]. In this study, we compared the antigenicity of N proteins of novel members of the hantavirus

**Table 3. Titeration of representative serum samples shown.**

| | | IFA titers of bat sera | | | | | |
| | Antigens | ZFB06-11 | ZFB07-14 | ZFB10-41 | ZFB10-44 | ZFB11-40 | ZFB14-151 |
|---|---|---|---|---|---|---|---|
| Bat-borne | BRNV | - | - | - | 4800 | 19200 | 19200 |
| | DKGV | 600 | - | - | - | - | - |
| | QNZV | - | - | - | - | - | - |
| | ROBV | - | 19200 | - | - | - | 600 |
| | XSV | 19200 | - | - | - | - | - |
| Shrew-borne | ASAV | - | - | - | - | - | - |
| | ALTV | - | - | 9600 | - | - | - |
| | CBNV | - | - | - | - | 4800 | - |
| | NVAV | - | - | - | - | - | - |
| | SWSV | - | - | - | - | - | - |
| | TPMV | - | - | 19200 | - | - | - |

-; <300

recombinant protein expression system in a wet laboratory. No strong cross-reactivity was detected among the rN antigens of the 11 viruses; however, low cross-reactivity was found between highly immunized serum and the rN antigens. Common but not dominant epitopes might induce antibodies at a low frequency leading to low-level cross-reactivity. However, by comparing their homologous titer values, it appeared that these rNs had different antigenicity. Therefore, we used all 11 rN antigens for serological screening. Two rNs of ROBV and QZNV appeared to have more similar antigenicity than those of other antigen combinations. Accordingly, these two viruses belong to the same phylogenetic lineage (Fig 1A). However, none of

**Table 4. Seropositivities to Mix-antigen-Bat and Mix-antigen-Shrew followed by IFAT with individual antigens.**

| Species | Tested | Mix-antigen-Bat -positive | Individual antigens | | | | | Mix-antigen-Shrew-positive | Individual antigens | | | | | |
| | | | BRNV | DKGV | QZNV | ROBV | XSV | | ASAV | ALTV | CBNV | NVAV | SWSV | TPMV |
|---|---|---|---|---|---|---|---|---|---|---|---|---|---|---|
| *Eidolon helvum* | 986 | 156 (15.8%) | 78 | 13 | 65 | 27 | 35 | 195 (19.8%) | 63 | 44 | 45 | 0 | 0 | 52 |
| *Epomophorus crypturus* | 84 | 7 (8.5%) | 6 | 0 | 2 | 0 | 0 | 13 (15.9%) | 4 | 3 | 2 | 0 | 0 | 5 |
| *Hipposideros caffer* | 2 | 0 | - | - | - | - | - | 0 | - | - | - | - | - | - |
| *Macronycteris vittatus* | 182 | 0 | - | - | - | - | - | 3 (1.6%) | 0 | 1 | 0 | 0 | 0 | 1 |
| *Miniopterus fraterculus* | 38 | 0 | - | - | - | - | - | 0 | - | - | - | - | - | - |
| *Nycteris thebaica* | 93 | 4 (4.3%) | 0 | 0 | 0 | 4 | 0 | 1 (1.1%) | 0 | 1 | 0 | 0 | 0 | 0 |
| *Rhinolophus blasii* | 1 | 0 | - | - | - | - | - | 0 | - | - | - | - | - | - |
| *Rhinolophus swinnyi* | 4 | 0 | - | - | - | - | - | 0 | - | - | - | - | - | - |
| *Rhinolophus* sp. | 4 | 0 | - | - | - | - | - | 0 | - | - | - | - | - | - |
| *Rousettus aegyptiacus* | 370 | 34 (9.2%) | 13 | 2 | 13 | 7 | 6 | 94 (25.4%) | 23 | 41 | 1 | 0 | 0 | 28 |
| Grand Total | 1764 | 201 (11.4%) | 97 | 15 | 80 | 38 | 41 | 307 (17.4%) | 90 | 90 | 48 | 0 | 0 | 86 |

-; Not done

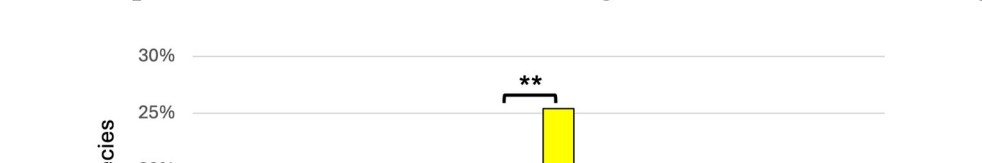

**A**    Comparison of reactivities to Mix-antigen-Bat and Shrew across species

**B**    Distribution of individual antigens across species

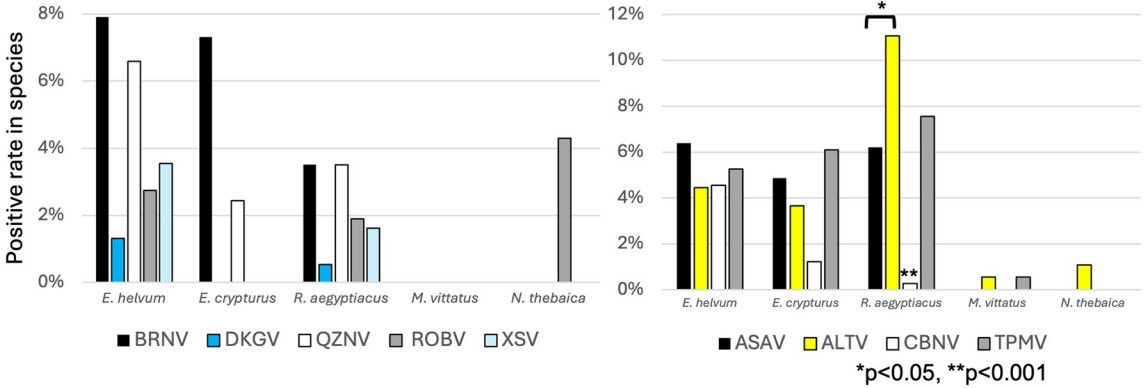

**Fig 6. Graphical distribution pattern of the novel system.** (A) Distribution pattern across bat species. *Eidolon helvum* showed the highest distribution for mixed-antigen-bat, whereas *Rousettus aegyptiacus* showed the highest distribution for mixed-antigen-shrew. (B) Similarly, the bar chart distribution of individual antigen reactivity across bat species revealed that Brno virus antibodies were dominant for *Eidolon helvum* followed by Altai virus antibodies for *Rousettus aegyptiacus*.

the bat serum samples showed cross-reactivity to ROBV and QZNV, suggesting that cross-reactivity between ROBV and QZNV might be found only in hyperimmune serum.

In this study, the rNs of ASAV and CBNV were reacted with the E5G6 monoclonal antibody, which recognizes an epitope located at amino acid positions 166–175 of the Hantaan virus N protein (S2 Table). This epitope sequence is conserved among rodent-borne orthohantaviruses [43]. The E5G6 epitope in rN proteins might not be immunodominant and antibodies against this epitope are unlikely to be induced during actual infection [44].

To efficiently screen the immune reactivity of bat serum samples for antigenically broad hantavirus antigens using IFAT, we attempted to express the rN proteins of 5 or 6 hantavirus species simultaneously. All the antigens were successfully expressed and detected using specific antisera. Samples positive for the mixed antigen were further tested by IFAT using rN antigens expressed individually in Vero E6 cells. In this step, some serum samples showed specific

reactivity to 2 or 3 viral antigens. Therefore, serial dilutions were performed and titrated against the respective antigens. IFA antibody titers above 4800 were required for western blot analysis of the positive bat serum samples.

The cut-off for cross-reactivity in laboratory-generated antisera according to this manuscript was estimated as 51200 titers. It was also clear that laboratory-generated sera titers were different from field antisera. While 4800 IFAT antibody titers of bat sera were also positive in Western blotting, 51200 IFAT titers from the laboratory-generated antisera were only cross-reactive in IFAT and not in Western blotting. Hence, a 51200 IFAT titer cut-off was used for the 11 antisera generated to estimate cross-reactivity. Generally, a 4-fold difference in antibody titer showed independent antigenicity. To evaluate true antigenic cross-reactivities, naturally infected animal sera are required. In this study, we did not detect cross-reactivity from bats sera captured in Zambia. Further careful study is needed to evaluate the cross-reactivity of bat-borne and shrew-borne hantavirus antigens. Regarding the reactivity of anti-bat IgG, it was designed to react with Bat IgG and as well as light chains common to other Bat immunoglobulins. The company showed confirmed reactivity to bat genus species *Pteropus vampirus*, *Desmodus rotundus*, *Eptesicus fuscus*, *Tadrida pumila*, *T. condylura*, *Hypsignathus monstrosus*, *Rosettus aegyptiacus*, *Epomorphus crypturus*, *Molossus species*, and *Phylostomus species*.

All bat serum samples were analyzed sequentially, and 11.4% were positive for mixed-antigen-bat. Among them, *Ei. helvum* showed the highest positivity rate (17.4%). Three species of fruit bats, *Ei. helvum*, *Ep. crypturus* and *R. aegyptiacus* showed higher seropositivity than insectivorous bats. However, the affinities of the secondary antibodies used in this study varied with the immunoglobulins of the different bat species. Therefore, we could not compare seropositivity among bat species. However, differences in positive patterns within the same species can be compared. *Ei. helvum* and *R. aegyptiacus* showed similar positivity patterns to the bat-borne hantavirus antigens. This may indicate that the opportunities for exposure to each virus were similar in the two fruit bat species. Thus, they may be terminal hosts infected by viral spillover, rather than reservoir hosts for these viruses. In contrast, the insectivorous bat *N. thebaica* was positive only for ROBV at low frequencies. The insectivorous bats may have been exposed to the virus via a different transmission route from the route by which the fruit bats were infected.

BRNV or related viruses might be an important hantavirus source in Zambian fruit bats. BRNV was originally reported in Europe; however, DKGV, QZNV, ROBV, and XSV have been detected in Asian and Oceanian countries. The number of positives of the QZNV antigen was similar to that of the BRNV (80 and 97, respectively). This seroprevalence suggests the possibility of the circulation of QZNV-related viruses in Zambia. Recently, a novel hantavirus, Buritiense mobatvirus, which belongs to the same lineage as QZNV and ROBV, was discovered in bats in Brazil [45]. QZNV- and ROBV-related viruses may also be distributed in Africa, and not only in Asia, Oceania, and South America.

The number of bats positive for mixed-antigen-shrew was larger than that of bats positive for mixed-antigen-bat. The Japanese shrew mole (*Urotrichus talpoides*) is native to Japan. Therefore, ASAV carried by Japanese shrew moles has never been detected outside Japan. Surprisingly, we detected a high seroprevalence of mixed-antigen-shrew in bats, with a particularly high positivity rate of the ASAV antigen in fruit bats. However, no bats serum samples were seropositive for the SWSV and NVAV antigens. According to phylogenetic analysis, ASAV is more closely related to SWSV than to ALTV and TPMV. Japanese shrew moles are not present in Zambia; therefore, alternative animal hosts must carry ASAV-related viruses.

*Rousettus* bats showed a different pattern against rN antigens of shrew-borne hantaviruses than that of the African fruit bat, *Eidolon helvum*. The Egyptian fruit bat, *Rousettus aegyptiacus*, showed a particularly high positivity rate for the ALTV antigen and a low positivity rate

for the CBNV antigen. The classification of ALTV has not yet been confirmed. ALTV was originally detected in *Sorex* shrews in Eurasia [46]. However, its genetic characteristics are distinct from those of other *Sorex*-borne viruses, and it appeared to be an intermediate virus between shrew-borne and bat-borne hantaviruses (Fig 1A). Therefore, the geographic distribution of ALTV and related viruses needs to be clarified. Further studies are needed to investigate the diversity of bat- and shrew-borne hantaviruses in Africa and to identify the reservoir hosts.

This retrospective study investigated the serological detection of bat- and shrew-borne hantaviruses circulating in Zambia. The presence of bats carrying viruses of zoonotic importance raises concerns about the possibility of human infection. Regarding human infectivity of shrew-borne hantaviruses, ALTV and TPMV seropositivity have been reported in humans in Sri Lanka and Thailand [27,31]. In addition, Imjin or Imjin-like viral infection has been detected in humans in China [35]. These findings suggest that shrew-borne viral infections have the potential to infect humans in several countries.

The serological strategy established in this study presented some information on the possible distribution of bat- and shrew-borne hantaviruses in Zambia. It is not known how bats harbor antibodies to shrew-borne viruses. Tree-climbing shrews may have contributed to their spread. Migratory fruit bats can cross highly contaminated areas during migration. It is also possible that shrew-borne viruses originated in bats. Further research is required to clarify the ecological dynamics of hantavirus transmission.

Previous reports have shown that bat-borne hantaviruses may have a worldwide distribution, except for Antarctica. Both partial and complete hantavirus sequences have been reported in Asia, Europe, and Africa. Further serological and molecular biological research is required to advance our understanding of the epidemiology of hantaviruses in bats, shrews, and humans in various regions.

This study therefore developed a serological system by expressing multiple antigens in a single transfection protocol to serve as a multiplex platform for testing five bat-borne and 6 shrew-borne antigens. Instead of testing all 1764 samples against individual antigens, which would have required 1,764 (5,292 in triplicate) wells expressing each antigen, the mixed antigen system reduces this to about 353 (1,059 in triplicate) and 294 (882 in triplicate) for the 5 bat- and 6 shrew-borne antigens respectively. This significantly reduces the number of wells while yielding equal results. Hence, this system helps to effectively pool all positive samples with fewer wells, followed by specific individual antigen analysis based on the number of positives detected. Apart from animal samples, this system can be used to screen human sera samples while using the required secondary antibodies. We, therefore, intend to apply this system in screening human sera samples in the future.

## Supporting information

**S1 Fig. Reactivities of bat sera to horseradish peroxidase conjugated anti-bat IgG, Protein A and Protein G The reaction of the conjugate with bat serum was examined as previously performed with rodent serum by Lee et al.** [39]. Briefly, bat serum was diluted with phosphate buffered saline at 1:1000 and absorbed to 96-well ELISA plate at 4C overnight. After blocking with bovine serum albumin, serially diluted conjugates were applied to wells. (TIFF)

**S1 Table. Comparison of individual antigens (A) and multiple positives (B) A: Pair-wise comparison between individual antigens; Fisher's least significant difference tests were performed between antigens.** No significant relationship between seropositivity to any antigens (DATAtab program). B: Correlation ratio between two antigens; A total of 138 bats showed positive reactions to multiple antigens. The relation between the reactivity of 138 was

examined considering the correlation between two antigens. The correlation ratio is shown in
S1B Table.
(XLSX)

**S2 Table. Amino acid sequences of the epitope and corresponding sites of monoclonal antibody E5/G6.** S2 Table Epitope sequence of monoclonal antibody E5/G6 on hantavirus N protein and comparison of reactivities of bat and shrew-borne hantaviruses.
(XLSX)

**S1 Data. Total IFA results and Seropositivity converted data.**
(XLSX)

## Author Contributions

**Conceptualization:** Rakiiya Sikatarii Sarii, Masahiro Kajihara, Ayato Takada, Kumiko Yoshimatsu.

**Data curation:** Rakiiya Sikatarii Sarii, Satoru Arai, Ayato Takada, Kumiko Yoshimatsu.

**Funding acquisition:** Satoru Arai, Ayato Takada, Kumiko Yoshimatsu.

**Investigation:** Rakiiya Sikatarii Sarii, Masahiro Kajihara, Zuoxing Wei, Sithumini M. W. Lokpathirage, Devinda S. Muthusinghe, Akina Mori-Kajihara, Katendi Changula, Yongjin Qiu, Joseph Ndebe, Bernard M. Hang'ombe, Fuka Kikuchi, Ai Hayashi, Motoi Suzuki, Hajime Kamiya, Satoru Arai.

**Methodology:** Akina Mori-Kajihara.

**Project administration:** Kumiko Yoshimatsu.

**Resources:** Masahiro Kajihara, Satoru Arai.

**Supervision:** Kumiko Yoshimatsu.

**Writing – original draft:** Rakiiya Sikatarii Sarii.

**Writing – review & editing:** Ayato Takada, Kumiko Yoshimatsu.

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
