## [Decision Letter · Decision Letter 0]

24 Sep 2024

Dear Dr. Yoshimatsu,

Thank you very much for submitting your manuscript "Development of a seroepidemiological tool for bat-borne and shrew-borne hantaviruses and its application using samples from Zambia" for consideration at PLOS Neglected Tropical Diseases. As with all papers reviewed by the journal, your manuscript was reviewed by members of the editorial board and by several independent reviewers. The reviewers appreciated the attention to an important topic. Based on the reviews, we are likely to accept this manuscript for publication, providing that you modify the manuscript according to the review recommendations. 

Sincerely,

David Safronetz, Ph.D.

Section Editor

David Safronetz

Section Editor

Reviewer's Responses to Questions

**Key Review Criteria Required for Acceptance?**

**Methods**

-Are the objectives of the study clearly articulated with a clear testable hypothesis stated?

-Is the study design appropriate to address the stated objectives?

-Is the population clearly described and appropriate for the hypothesis being tested?

-Is the sample size sufficient to ensure adequate power to address the hypothesis being tested?

-Were correct statistical analysis used to support conclusions?

-Are there concerns about ethical or regulatory requirements being met?

Reviewer #1: The section of the methods for the secondary Ab for bat IgG starting at line 204 through line 212 is the exact same as the section in the results from 289-297. The section of the results should be changed to provide a description of what was found, i.e. that the ODs were higher at greater dilutions for the Ab compared with protein A and G as shown in the figure. Also the ELISA should be described, at least in some detail to explain that reactivity against 3 bat species was tested and how those reactivities compared (for example in line 208 – reactive against what?). While this is partly addressed in the results and discussion later, what is the actual target of the commercial anti-bat IgG? Is there a predicted species reactivity of this Ab and how cross-reactive is it? While the Ab was “more reactive” against the 3 species tested, how broadly true is this? At lesser dilutions (around 1:400), protein G was equivalent, and possible more broadly cross-reactive than the anti-bat Ab. Would using protein G at 1:400 have allowed for better detection of bat Ab in your assays? This should be considered or at least commented on. Running a comparison would be worthwhile to see if any possible positive would be missed, especially if a broader range of species are to be tested in the future. 

With respect to the initial IFAT methodology (presented in Fig 2A) and used throughout, simply having the negative control consist of untransfected VeroE6 cells and unimmunized serum is not sufficient. Using Fig 2A as an example, the results presented are clear, however additional controls should be added. This includes serum from unimmunized mice bound to transfected cells (all antigens) to show that there is no non-specific binding leading to fluorescence in your assay. And sera from immunized mice bound to untranfected cells will show that there is no non-specific binding of immune sera to VeroE6 cells that do not express hantavirus N antigens. Also, the use of the monoclonal as a positive control against these antigens would be beneficial. As the data are currently presented, these cannot be ruled out.

Reviewer #2: No concerns

Reviewer #3: There needs to be information on group sizes, repeats, how many times things were done. I have no idea what the n is for any of the assays. Based on the western blots I assume the n=4 but this needs to be stated.

The source of materials needs to be harmonized e.g. (company name) vs (company name, city, state, country) throughout. 

Some methods are missing e.g. how they performed the Nus tag western blot as opposed to the bat sera western blot.

**Results**

-Does the analysis presented match the analysis plan?

-Are the results clearly and completely presented?

-Are the figures (Tables, Images) of sufficient quality for clarity?

Reviewer #1: Figure 2: there should be labels for each of the individual antigens added to Fig 2A.

Reviewer #2: (1) Figure 1: Parts B and C could be considered for Supplemental Material. Figure 1 could then be expanded by a comparison among the N proteins (identity percentage). 

(4) Figure 2: The authors should label the different IFAs in part A. For part B, the authors should indicate the location of the expressed rN. Please explain the multiple bands of higher and lower molecular weight reacting with the antibodies - degradation or processing products.

(5) Table 1: The authors should define and provide the rational for their cross-reactivity cut-off.

(6) Figure 3: As a control the authors should have considered transfection with a plasmid expressing an unrelated protein or just ‘empty’ plasmid DNA rather than non-transfected cells. 

(7) Serological screening: Are the 107 sera positive to both mixed antigens true co-infections. If yes, is there a pattern or a random distribution regarding individual antigens? This should be better described. 

(8) Figures 4 & 5: These figures may need some redesigning as they interrupt the text flow. 

(9) Make sure that the figure legends contain all important information.

Reviewer #3: Tables have to be updated as the math or the titled does not add up e.g. Table 2 Eidolon helvum row mix-antigen-shrew needs to state that the positive to both is a part of positive to either, not in addition to the 270.

Figure 4 and 5 need to be made more consistent with respect to how many sections they have and the sample numbers need to be modified to not be what they are called in the author's spreadsheets.

The final section has no title and should be combined with the single rN expression section.

Not sure how the statistical analysis is done with respect to how the p values were calculated and there are no error bars.

sample numbers have to be stated on figures.

**Conclusions**

-Are the conclusions supported by the data presented?

-Are the limitations of analysis clearly described?

-Do the authors discuss how these data can be helpful to advance our understanding of the topic under study?

-Is public health relevance addressed?

Reviewer #1: (No Response)

Reviewer #2: No concerns

Reviewer #3: The the conclusions that this is a useful tool for seroepidemiological studies is well supported, however the statistical analysis appears to be weak likely due to the fact that the sample for this study wasn't intended for this tool and may not have the required numbers for appropriate statistical power. This is expected as this was an opportunistic use of already available samples that were collected for another purpose. This in no way invalidates the publishing of the tool which is the real strength of this paper.

**Editorial and Data Presentation Modifications?**

Reviewer #1: (No Response)

Reviewer #2: minor revision

Reviewer #3: I would like to see another paragraph that discusses the actual tool in the discussion as this is the big development of this paper as far as I see it. 

The figures need to be made more consistent and the table needs to be removed from figure 4 and set up as a standalone table.

**Summary and General Comments**

Reviewer #1: (No Response)

Reviewer #2: The manuscript by Sikatarii and colleagues describes the development of seroepidemiologic tools for bat-borne and shrew-borne hantaviruses. Recombinant nucleoproteins from different bat- and shrew-borne hantaviruses were expressed in cells and used to screen bat sera from Zambia. The authors conclude that bats in Zambia are likely exposed to both types of hantaviruses.

General Comments:

(1) The detection of serological responses in bats against shrew-borne hantaviruses is surprising and likely cannot be easily explained. Is there evidence others than serology that would support this finding such as genome detection by RT-PCR or virus isolation? The findings should be discussed with caution here. 

(2) An ELISA platform would likely be easier and simpler than an IFAT/westernblot. The authors should provide their rational for their choice and discuss the ELISA option. 

(3) The low level of cross-reactivity among the different hantavirus N proteins is surprising. Why is that? What is the sequence identity of the different N proteins in percentage? This should be described/discussed, and the comparison be presented as a table in Supplemental Material.

Reviewer #3: The manuscript “Development of a seroepidemiological tool for bat-borne and shrew-borne hantaviruses and its application using samples from Zambia” by Sikatarii et al. is an interesting paper that describes the development of a serological testing algorithm and suggests that “shrew-borne” hantaviruses more commonly infect bats than “bat-borne” hantaviruses. The sampling appears to be opportunistic from previously available samples so is lacking the numbers needed for extensive analysis (this is not necessarily a negative but the bat capture study does not appear to have the statistical power for confident modeling) but is convincing to show the utility of the seroepidemiological tool. There are several confusing or missing information in the paper which needs to be clarified prior to submission (see specific comments below).

PLOS authors have the option to publish the peer review history of their article (what does this mean?). If published, this will include your full peer review and any attached files.

Reviewer #1: No

Reviewer #2: No

Reviewer #3: No

Figure Files:

Data Requirements:

Reproducibility:

References

---

## [Editor Report · Decision Letter 1]

30 Oct 2024

Dear Dr. Yoshimatsu,

We are pleased to inform you that your manuscript 'Development of a seroepidemiological tool for bat-borne and shrew-borne hantaviruses and its application using samples from Zambia' has been provisionally accepted for publication in PLOS Neglected Tropical Diseases.

Best regards,

David Safronetz, Ph.D.

Section Editor

David Safronetz

Section Editor

Shaden Kamhawi

co-Editor-in-Chief

Paul Brindley

co-Editor-in-Chief

---

## [Editor Report · Acceptance letter]

14 Nov 2024

Dear Dr. Yoshimatsu,

We are delighted to inform you that your manuscript, "Development of a seroepidemiological tool for bat-borne and shrew-borne hantaviruses and its application using samples from Zambia," has been formally accepted for publication in PLOS Neglected Tropical Diseases.

Best regards,

Shaden Kamhawi

co-Editor-in-Chief

Paul Brindley

co-Editor-in-Chief
